# Identification of the WRKY Gene Family and Characterization of Stress-Responsive Genes in *Taraxacum kok-saghyz* Rodin

**DOI:** 10.3390/ijms231810270

**Published:** 2022-09-07

**Authors:** Yifeng Cheng, Jinxue Luo, Hao Li, Feng Wei, Yuqi Zhang, Haiyang Jiang, Xiaojian Peng

**Affiliations:** National Engineering Laboratory of Crop Stress Resistance Breeding, School of Life Sciences, Anhui Agricultural University, Hefei 230036, China

**Keywords:** WRKY, *Taraxacum kok-saghyz* Rodin (TKS), genome-wide analysis, expression patterns, abiotic stress, hormonal treatments

## Abstract

WRKY transcription factors present unusual research value because of their critical roles in plant physiological processes and stress responses. *Taraxacum kok-saghyz* Rodin (TKS) is a perennial herb of dandelion in the Asteraceae family. However, the research on TKS WRKY TFs is limited. In this study, 72 TKS WRKY TFs were identified and named. Further comparison of the core motifs and the structure of the WRKY motif was analyzed. These TFs were divided into three groups through phylogenetic analysis. Genes in the same group of TkWRKY usually exhibit a similar exon-intron structure and motif composition. In addition, virtually all the TKS WRKY genes contained several *cis*-elements related to stress response. Expression profiling of the TkWRKY genes was assessed using transcriptome data sets and Real-Time RT-PCR data in tissues during physiological development, under abiotic stress and hormonal treatments. For instance, the *TkWRKY18*, *TkWRKY23*, and *TkWRKY38* genes were significantly upregulated during cold stress, whereas the *TkWRKY21* gene was upregulated under heat-stress conditions. These results could provide a basis for further studies on the function of the TKS WRKY gene family and genetic amelioration of TKS germplasm.

## 1. Introduction

The essence of TFs is DNA-binding proteins, which can specifically interact with upstream/downstream genes to regulate gene transcription [1]. This is incredibly plant specific, sometimes even in high proportion, for WRKY TFs in higher plants [2,3]. WRKY domains consisted of approximately 60 amino acid residues, mainly comprising a conserved domain of WRKYGQK at the N-terminus, followed by a zinc finger motif C_2_H_2_ (C-X_4–5_-C-X_22–23_-H-X-H) or C_2_HC (C-X_7_-C-X_23_-H-X-C) at the C terminal [4,5,6]. WRKY proteins were divided into three groups determined by the WRKY domains. All proteins containing two WRKY domains were divided into Group I, whereas the remaining two groups contain only one. The slight difference between the remaining two groups was the type of their zinc finger motifs. Group II contained a C_2_H_2_ type motif whereas the C_2_HC type belonged to Group III [7].

Now, a growing stream of research suggests that the WRKY gene family plays important roles in plant development and substance metabolism. Many WRKY proteins have been certified in the plant kingdom, which are mainly related to seed dormancy and germination [8,9], embryogenesis [10,11], trichome development [12,13], pollen development [14,15], root development [16], modulation of flowering time [17], leaf senescence [18], plant nutrient utilization [19], and biosynthesis of secondary metabolites [20]. Although they play important roles in these situations, more important are their biological functions and molecular regulatory mechanisms in plant responses to biological and abiotic stresses [21]. The growth of the knockout lines of the *ATWRKY4* gene is significantly inhibited under salinity or Me-JA stress. The *ATWRKY3/WRKY4* knockout transgenic plants also exhibit an increase in relative electrolyte leakage (REL) and a significant decrease in antioxidant enzyme activity [22]. For overexpression of rice transcriptional repressor *OsWRKY76*, which is expressed in rice plants, the tolerance to cold stress is increased [23]. The tolerance of transgenic *Arabidopsis* to high temperature is increased by the overexpression of *ZmWRKY106* involved the ABA signaling pathway, and it reduces the reactive oxygen species in transgenic lines by enhancing the activity of superoxide dismutase and catalase under drought stress [24]. The four wheat WRKY TFs, including *TaWRKY142*, *TaWRKY112*, *TaWRKY92*, and *TaWRKY49*, are usually upregulated in hybrid necrosis and hybrid chlorosis. The expression of either *TaWRKY142*, *TaWRKY112*, *TaWRKY92*, or *TaWRKY49* will increase the tolerance to salinity and osmotic stress. These proteins serve as a comprehensive hub of multiple stress-signaling pathways, and they play a critical role in hybrid necrosis and hybrid chlorosis induced by autoimmune response [25]. *SlWRKY8* can activate W-box-dependent transcription in tomato, which plays a positive regulatory role in immunity against pathogen infection and response to salinity stress [26].

TKS is considered as an important potential alternative source for the production of natural inulin and rubber [27,28,29]. Pictures of TKS1151 at different stages are as follows (Figure 1). TKS roots contain latex and can produce rubber, which is used to make general rubber products. TKS leaves are rich in chicory acid, which has a potential antioxidant application and can be used as a nutritional component in Chinese herbal medicine and healthy feed production [30]. Susceptibility to allergic reactions and the spread of fungal pathogens of South American leaf blight disease have both reduced the production of *Hevea brasiliensis*, so TKS is of strategic and economic importance as a potential alternative source of natural rubber [31]. Compared with other rubber-producing plants, TKS has the ideal agronomic characteristics and can be used as an annual crop for high-efficiency production, including rapid development and high biomass production. In addition, TKS has several advantages, such as a wide growth range, high content of natural rubber, rubber with no protein allergy, relatively simple genome, and easy genetic transformation and gene editing [32,33]. Therefore, from the perspective of genomics, it proposes suggestions for further study of rubber plants. This provides a good idea and an important research direction for the cultivation of high-quality TKS.

The WRKY gene family has been identified and studied in many plants except TKS. With the publication of the draft genome, it becomes possible to identify the WRKY TFs of TKS at the genome-wide scale. In this study, 72 TKS WRKY genes were identified and divided into three groups. Further comprehensive analysis was made from the exon-intron structure, motif composition, phylogeny, and *cis*-acting elements. Some genes that may play critical roles in abiotic stress and plant hormone signaling pathways through global expression analysis were identified. This study aims to provide new ideas for the application and popularization of WRKY genes in TKS, and some genes of this gene family may be used for TKS quality improvement and become resources for resistance breeding in crops.

## 2. Result

### 2.1. Identification of the WRKY Protein Family in TKS

To identify the WRKY proteins in TKS, a BLASTP search was performed using the conserved WRKY domain in *Arabidopsis*. The putative WRKY proteins were validated by Pfam and SMART programs. At the same time, 72 putative WRKY proteins were identified using the InterProScan Database with the hidden Markov model (PF03106). According to the genome database information, they were named TkWRKY1 to TkWRKY72, corresponding to the size of the accession number.

The physicochemical properties of the TkWRKY proteins were analyzed. Among the 72 TkWRKY proteins, TkWRKY8 (67 aa) was the smallest protein whereas the largest protein was TkWRKY23 (676 aa). The MW of the proteins ranged from 7.74 kDa to 75.09 kDa. The pI ranged from 4.88 (TkWRKY54) to 10.07 (TkWRKY8), and half of the proteins had isoelectric points greater than 7. The instability index and grand average of hydropathicity (GRAVY) of all the proteins were greater than 40 and less than 0, respectively. The Aliphatic index had a maximum of 85.2 (TkWRKY70) and a minimum of 38.09 (TkWRKY9). The subcellular location prediction results indicated that 71 TkWRKY proteins were located in the nucleus region and only 1 TkWRKY protein was located in the mitochondria (Appendix A). In conclusion, TKS WRKY proteins may be instable and hydrophilic.

### 2.2. Multiple Sequence Alignment, Structural Analysis of the WRKY Variants, and Phylogenetic Analysis of the TkWRKY Genes

The phylogenetic relationship of the WRKY protein domain in TKS was studied by multiple sequence alignment. In total, 70 amino acid domains of the TKS WRKY protein domain were intercepted, and we then selected the WRKY domain of 7 different *Arabidopsis* WRKY proteins (ATWRKY58, ATWRKY18, ATWRKY61, ATWRKY13, ATWRKY17, ATWRKY65, and ATWRKY46) from each group or subgroup as representatives for further comparison. The sequence in the WRKY domain was highly conserved, whereas four TkWRKY proteins (TkWRKY6, TkWRKY24, TkWRKY58, and TkWRKY41) were variants in the WRKY domain (Figure 2). Many WRKY domain variants have been observed in plant species such as Camelina and legumes [34,35]. The core motif WRKY of the variants was replaced by other amino acids. A zinc-finger structure unlikely formed due to the lack of specific cysteine or histidine. However, the proportion of these variants was very small in the TKS WRKY protein.

Phylogenetic analysis of *Taraxacum kok-saghyz* Rodin and *Arabidopsis* was conducted based on the WRKY domain (Figure 3). TKS WRKY domains were divided into three main groups, namely, Groups I, II, and III, where the second group contained five subgroups [5,6]. Sixteen TKS WRKY genes (*TkWRKY2*, *TkWRKY11*, *TkWRKY13*, *TkWRKY18*, *TkWRKY25*, *TkWRKY29*, *TkWRKY31*, *TkWRKY35*, *TkWRKY36*, *TkWRKY38*, *TkWRKY42*, *TkWRKY44*, *TkWRKY47*, *TkWRKY48*, *TkWRKY64*, and *TkWRKY67*) with two WRKY domains belonged to Group I, which comprised a zinc-finger motif of C_2_H_2_. Based on the phylogenetic relationship between TKS and *Arabidopsis* expressed by the tree branch graph of the evolutionary tree, the other 43 TKS *WRKY* genes with the C_2_H_2_ types were assigned to Group II, which contained only one WRKY domain. The 43 Group II TKS WRKY genes were distributed into five subgroups: Group II-a (two: *TkWRKY52* and *TkWRKY71*); Group II-b (eleven: *TkWRKY5*, *TkWRKY12*, *TkWRKY14*, *TkWRKY17*, *TkWRKY19*, *TkWRKY21*, *TkWRKY26*, *TkWRKY34*, *TkWRKY40*, *TkWRKY53*, and *TkWRKY69*); Group II-c (seventeen: *TkWRKY3*, *TkWRKY4*, *TkWRKY16*, *TkWRKY20*, *TkWRKY23*, *TkWRKY24*, *TkWRKY28*, *TkWRKY30*, *TkWRKY41*, *TkWRKY46*, *TkWRKY49*, *TkWRKY55*, *TkWRKY57*, *TkWRKY58*, *TkWRKY61*, *TkWRKY65*, and *TkWRKY66*); Group II-d (five: *TkWRKY7*, *TkWRKY22*, *TkWRKY51*, *TkWRKY56*, and *TkWRKY63*); and Group II-e (eight: *TkWRKY1*, *TkWRKY6*, *TkWRKY8*, *TkWRKY9*, *TkWRKY37*, *TkWRKY50*, *TkWRKY62*, and *TkWRKY68*). Unlike the other two groups, Group III genes contained a C_2_HC type and included twelve WRKY genes in TKS (*TkWRKY10*, *TkWRKY15*, *TkWRKY27*, *TkWRKY32*, *TkWRKY33*, *TkWRKY39*, *TkWRKY43*, *TkWRKY45*, *TkWRKY59*, *TkWRKY60*, *TkWRKY70*, and *TkWRKY72*).

The phylogenetic tree is divided into groups I, II-a, II-b, II-c, II-d, II-e, and III with green, dark yellow, dark blue, orange, red, yellow and blue marking the clusters, respectively. 

### 2.3. Gene Structure and MOTIF Composition of TKS WRKY Gene Family

Since the diversity of the gene structure drives the evolution of multi-gene families, the exon-intron structure of the identified 72 TkWRKY genes were analyzed (Figure 4). It is obvious that Group I contained 2–5 introns, Group II-a contained 3–4 introns, Group II-b contained 2–5 introns, and Group II-c contained 0–5 introns, half of which contained three introns. Group II-d and Group II-e all contained 2 introns, excluding *TkWRKY1*. Group III varied from 1 to 4 introns (Figure 4b). There were significant differences in the number of introns and exons in the different groups, indicating that the TkWRKY gene structure was relatively complex and may have undergone domain shuffling after genome replication. Genes in the same group usually had similar structures. For example, all TkWRKY genes in Group II-d contained 3 exons and 2 introns [36,37].

To further study the structure and phylogenetic relationships of TKS WRKYs, the conserved motifs were obtained, and schematic diagrams of all TkWRKY proteins were constructed according to the motif analysis results. WRKY motifs in the same group contained similar structures. Motif 1 and 4 were WRKY domains; Motif 4 was only present in Group I whereas Motif 1 was distributed the whole groups. Motif 10 was unique to Groups II-a and II-b, whereas Motif 8 to Group II-b and Motif 15 to Group II-c (Appendix A). Clustered TkWRKY pairs, such as *TkWRKY25/29* and *TkWRKY10/43*, exhibited a highly similar motif distribution (Figure 4c). Although some proteins contained unknown functions, the motif structure of most TkWRKY proteins in the same subgroup was similar, indicating that the protein structure of this family was conserved in a specific subfamily. All in all, the similar gene structure, conservative motif arrangement, and phylogenetic analysis of the same group of proteins can be used as an important basis for protein classification.

### 2.4. Evolutionary Analysis of Group III WRKY Genes in TKS and Several Different Species

The WRKY III gene in plants originated after the differentiation of monocots and dicots, and it seemed to play a critical role in plant adaptation and evolution [38]. To study the evolution of the TKS Group III WRKY gene, MEGA7 was used to construct four monocotyledonous plants (rice, maize, *Brachypodium distachyon*, and pineapple) and five dicotyledonous plants (*Arabidopsis*, grapes, peaches, poplars, and *Taraxacum kok-saghyz* Rodin) of Group III WRKY protein phylogenetic trees. All the WRKY family members of Group III was divided into 8 clades (Figure 5). The species that aggregated in the same clade were closely related. Most proteins in dicotyledons were clustered on Clade 1, 4, and 7, whereas most proteins in monocotyledons were clustered on Clade 1, 2, 3, 5, 6, and 8, which strongly indicated that the evolutionary relationships among the WRKY members of the same clade were similar. Clade 1 contained proteins from all nine species, suggesting that these proteins may be orthologs from a single ancestor gene. Motif 1 and 11 were WRKY domains, and Motif 2 was a C_2_HC zinc-finger motif. Interestingly, Motif 1 and 2 were found in all proteins, indicating these two motifs may play different functions in different species. Motif 15 and 16 were only found in the dicotyledonous plant, *Arabidopsis*, where Motif 3 was only found in TKS and Motif 8 was only found in dicotyledonous plants and pineapples (Figure 5 and Appendix A). In general, the conserved motif composition and similar gene structure of the WRKY members in the same group, as well as the results of the phylogenetic analysis, strongly supported the reliability of the population classification.

### 2.5. Stress-Related Cis-Elements in TkWRKY Promoter Regions

To further study the potential regulatory mechanism of WRKY TFs involved in stress response, 12 representative *cis*-elements were selected for quantitative and functional analysis (Figure 6). At the same time, according to their functions, they were divided into three categories, namely, related to environmental stress, hormone response, and development; the number of each *cis*-element is counted in Appendix A.

In the current study, all TkWRKY TFs contained 2–9 *cis*-elements associated with stress response except *TkWRKY43* and *TkWRKY44*, probably because their protein sequences were too short. Many TkWRKY promoter regions contained identified elements related to hormone regulation, such as ABRE, CGTCA−motif, GARE−motif, TCA−element, and TGA−element. In total, 63 TkWRKYs (87.5%) contained one or more ABREs, indicating that there was a potential abscisic acid reaction under pressure [39]. CGTCA motifs were involved in the MeJA response in 56 TkWRKYs (77.8%). TCA−element, TGA−element, and GARE−motif were found in 25 (34.7%), 26 (36.1%), and 9 (12.5%) TKWRKYs, which were the response elements of salicylic acid, auxin and gibberellin, respectively. Among the environmental stress-related elements in the TkWRKY gene family, about half of the genes included WUN−motif, W−box, and TC−rich repeats. They play critical roles in plant defense and stress responsiveness and response to mechanical damage, respectively. In addition, about half of the genes included LTR and MBS, which contribute significantly to plant responses to low temperature and drought induction, respectively. Significantly, the only development-related element was circadian, which was associated with responsiveness to meristem development and controls circadian rhythms [40]. These results play a decisive role in the profound cognition of stress-related *cis*-elements in TkWRKY.

### 2.6. Expression Profiling of TKS WRKY Genes with RNA-Seq

To determine the expression pattern of 72 TkWRKY genes at different developmental stages in TKS tissues, we conducted a hierarchical clustering heat map with the public transcriptome data obtained from the NGDC Genome Warehouse (Appendix A). Twelve tissues were obtained from the database, including flowers (FL), latex (LA), peduncle (PE), seed (SE), mature leaf (ML), mature lateral root (MLR), mature main root (MMR), mature stem (MS), young leaf (YL), young lateral root (YLR), young main root (YMR), and young stem (YS) [29]. Among the 72 TkWRKY genes, the expression of *TkWRKY61* in all detected samples was almost 0, which seemed to be a pseudogene or a special spatiotemporal expression pattern that was not detected in our library. In total, 55 TkWRKY genes were found to be expressed in all 12 tissues (FPKM > 0), and 20 of them showed constitutive expression, such as *TkWRKY1*, *TkWRKY35*, *TkWRKY6*, *TkWRKY56*, *TkWRKY7*, *TkWRKY60*, *TkWRKY48*, and *TkWRKY67* (Figure 7a). Obviously, the expression pattern of the TkWRKY genes in TKS was tissue specific. For instance, *TkWRKY34* was induced only in young lateral root, whereas *TkWRKY45* in leaves. The root and stem tissues were gathered into a large clade. Leaf, seed, and pedicels were clustered into one clade whereas the pedicel is more closely related to stem. Flower and latex belonged to separate branches, but latex was more closely related to roots. In general, the expression level of reproductive tissue (flower and seed) in TKS was generally lower than that of vegetative tissue (leaf, root, peduncle, and stem). The expression levels of many TkWRKY genes in mature lateral roots, mature stems, young lateral roots, young main roots, and young stems were higher than that in other tissues (Figure 7b). For example, *TkWRKY3*, *TkWRKY49*, *TkWRKY59*, *TkWRKY37*, *TkWRKY65*, *TkWRKY18*, *TkWRKY62*, *TkWRKY71*, *TkWRKY64*, *TkWRKY50* and *TkWRKY17*. *TkWRKY38*, *TkWRKY63*, *TkWRKY64*, and *TkWRKY71* exhibited high transcription abundance in almost all tissues and clustered into a group, whereas *TkWRKY14*, *TkWRKY20*, *TkWRKY32*, *TkWRKY45*, *TkWRKY53*, *TkWRKY54*, and *TkWRKY55* exhibited extremely low expression levels in all tested tissues. Interestingly, the expression level of several genes showed significant trends at different developmental stages. For instance, with the maturity of the main root, lateral roots, and stems, *TkWRKY72*, *TkWRKY36*, and *TkWRKY18* were depressed during the process. The expression level of *TkWRKY71* was lower in the seed sample, but higher in young and mature tissues. *TkWRKY27* expression increased with stem maturation (Figure 7a).

### 2.7. Expression Patterns of TKS WRKY Genes in Response to Different Treatments

To further investigated the changes in the TkWRKY gene expression levels under different abiotic stress and hormone treatments, 14 genes were carefully selected with high expression levels in the expression profile or specific expression in tissues. Real-time RT-PCR was further performed to detect the expression patterns under different treatments. In general, there were multiple TkWRKY genes that can be induced simultaneously by a single treatment. Eight TkWRKY genes (*TkWRKY4*, *TkWRKY10*, *TkWRKY13*, *TkWRKY18*, *TkWRKY23*, *TkWRKY28*, *TkWRKY38*, and *TkWRKY71*) were induced by cold stress (Figure 8a). Seven genes (*TkWRKY4*, *TkWRKY6*, *TkWRKY13*, *TkWRKY21*, *TkWRKY23*, *TkWRKY27*, and *TkWRKY38*) were induced by heat stress (Figure 8b). Five genes (*TkWRKY6*, *TkWRKY10*, *TkWRKY18*, *TkWRKY27*, and *TkWRKY39*) were induced by NaCl treatment (Figure 8c). Five genes (*TkWRKY6*, *TkWRKY10*, *TkWRKY18*, *TkWRKY27*, and *TkWRKY39*) were induced by PEG treatment (Figure 8d), whereas three genes (*TkWRKY10*, *TkWRKY13*, and *TkWRKY21*) were induced by ABA treatment (Figure 9a). Three genes (*TkWRKY18*, *TkWRKY38*, and *TkWRKY27*) were induced by MeJA treatment (Figure 9b). Three genes (*TkWRKY18*, *TkWRKY38*, and *TkWRKY71*) were induced by SA treatment (Figure 9c). Conversely, a single TkWRKY can be significantly induced/suppressed under a variety of treatments. For instance, both *TkWRKY18* and *TkWRKY38* belonged to the same group and significantly responded to MeJA treatment, SA, and cold treatments, suggesting that genes from the same populations may have similar functions. Moreover, *TkWRKY21* was induced by ABA treatment and heat stress. *TkWRKY10* was induced by NaCl, PEG, and ABA treatments. *TkWRKY71* was induced by cold stress and SA treatment. Interestingly, the transcript levels of several TkWRKYs, such as *TkWRKY21*, *TkWRKY23*, and *TkWRKY39*, were repressed intensively by the SA and MeJA treatments. This demonstrated that they may be involved in negative regulation of the SA pathway. Several genes exhibited the opposite or identical expression patterns under different treatments. For example, *TkWRKY23* was significantly upregulated by heat stress, whereas it was repressed by SA treatment. The expression level of *TkWRKY38* was downregulated under ABA and PEG treatments, but it was upregulated after the cold and heat stresses. Both *TkWRKY21* and *TkWRKY23* were repressed by the MeJA and SA treatments. Furthermore, the expression level in several genes fluctuated under a complete single-treatment process. For instance, the expression of *TkWRKY13* was rapidly upregulated at 9 h after cold stress but repressed at 16 and 24 h. Similarly, *TkWRKY23* was induced by heat and cold stresses, but the turning points were the 16 h and 9 h timepoints, respectively. *TkWRKY18* under heat stress was repressed at 3 and 6 h, but induced at 9 and 16 h, and then repressed at 24 h. Notability, *TkWRKY27* was induced by NaCl, PEG, and SA treatments at the 3 h timepoints and then repressed over the rest of the process. It was indicated that this gene can quickly respond to various abiotic stresses and hormone treatments. These results primarily revealed the diverse mechanisms controlling TKS WRKY gene responses to various intensities of abiotic stresses and hormonal treatments (Figure 8 and Figure 9).

## 3. Discussion

As reported, genome-wide- analysis of the WRKY gene family has been studied for multiple plants, including maize [41], *Helianthus annuus* L. [42], sweet Osmanthus [43], white pear [44], upland cotton [45], and switchgrass [46]. In this study, 72 WRKY genes were identified based on TKS genome sequences.

Although the WRKY motif is almost conserved in many plants with the WRKYGQK domain, there were many kinds of variants in the WRKY domain reported in previous research. Grapes have been reported to have two variants (WRKYGKK and WKKYGQK) [47] and eggplant has seven variants: WIKYGQK, WRKYGEK, WRKYGKK, WKKYEIG, WRKYGQN, WHKFGQK, and WDKFGQK [48]. Moreover, five variants have been observed in poplars (WRKYGKK, FWRKYGQK, WKKYGQK, WRKYGRK, and WRKYGEK) [36] and 1 in cucumber (WRKYGKK) [49]. In this study, domain recognition of the TKS WRKY domains revealed that three variants with WRKYGKK belonged to Group II-c (TkWRKY24, TkWRKY41, and TkWRKY58) and one (TkWRKY6) with WKKYGEK belonged to Group II-e (Figure 2). Furthermore, the uniqueness of the TkWRKY6 domain and the group to which it belonged revealed that TkWRKY6 may possess a different biological function. Interestingly, TKS contained three WRKYGKK variants in Group II-c and similar variants were also found in Group II-c of *Helianthus annuus* L. [42]. It seemed that this mutation was indeed common in Asteraceae. The existence of the WRKYGQK motif variants may increase the diversity of WRKY protein binding to target genes. Perhaps more binding elements will bind to proteins with more mutants, not just W-box elements [5].

Domain loss is a normal phenomenon, which is considered to be the divergent power of the gene family expansion [4,50]. In the current study, the Group I proteins in TKS all contained two WRKY domains with no domain loss event found, whereas some genes in Group II-c, II-e, and III had domain deletions, indicating that these groups performed different functions during their evolution. In total, 72 WRKY TFs were identified from 46,731 TKS annotated genes (Figure 2). Compared with the number of genes of *Helianthus annuus* L. (90) and rice (100), the size of the WRKY family in TKS is small. Similar to other WRKY gene families, such as *Arabidopsis* and *Helianthus annuus* L., the WRKY genes in Group II-a account for a small proportion and seemed to play various roles in dealing with abiotic stress [51,52]. Interestingly, phylogenetic and comparative genomics studies have indicated that Group II-a was the last to evolve and appears to be derived from Group II-b [6]. Although the whole-genome sequence of TKS was obtained, the information about the chromosomal location of the TkWRKY genes is still unknown; therefore, analysis of the synteny and gene-duplication events of this species could not be performed in this work.

Gene structure and motif analysis of the TKS WRKY gene family show that genes on the same branch usually have a similar structure or domain composition. These results can be used as an important basis for WRKY protein classification (Figure 4). Group III WRKY genes are considered to be the most advanced, dynamic, and adaptable gene group [38]. The diversity in the number of WRKY gene families is due to the difference in Group III in different species. For instance, TKS WRKY III contains 12 genes, fewer than rice or maize, but more than pineapple or grape, so the number of WRKY III may represent the total number of genes in the family to some extent (Figure 5). Group III WRKY genes formed monocotyledonous and dicotyledonous plant-specific subbranches, indicating that the WRKY gene evolved independently after monocotyledonous plant-dicotyledonous plant division. Therefore, Group III WRKY genes not only originated from the differentiation of monocotyledons and dicotyledons, but also evolved independently after the division of them, which proved their advanced nature and strong adaptability [38].

WRKY stress-related *cis*-elements play critical roles in several plant processes under ever-changing environmental conditions by regulating the plant complex defense signal transduction pathways [53]. WRKY TFs interacted with the *cis*-elements downstream of gene promoters to regulate the expression of target gene and caused a series of responses, consequently enhancing the resistance to stress of plants [49,54,55,56,57]. WRKY transcription factors can recognize and bind w-box elements to regulate the expression of downstream genes [4]. In this study, the 12 *cis*-elements were ABRE (*cis*-acting element involved in the abscisic acid responsiveness), CGTCA−motif (*cis*-acting regulatory element involved in the MeJA-responsiveness), GARE−motif (gibberellin-responsive element), LTR (*cis*-acting element involved in low-temperature responsiveness), MBS (MYB binding site involved in drought-inducibility), MBSI (MYB binding site involved in flavonoid biosynthetic genes regulation), TC−rich repeats (*cis*-acting element involved in defense and stress responsiveness), TCA−element (auxin-responsive element), TGA−element (auxin-responsive element), W−box (WRKY transcription factor binding site in defense responses), WUN−motif (mechanical damage response element), and circadian (*cis*-acting regulatory element involved in circadian control) (Figure 6). In total, 32 TKS WRKY genes contained W−box elements, which may interact with other WRKY transcription factors. The TKS WRKY promoter also contained several hormone-responsive elements, such as ABRE, CGTCA−motif, GARE−motif, TCA−element, and TGA−element, suggesting that they may be involved in a wide variety of hormone-mediated signaling pathways. The MYB-binding sites MBS and MBSI are also present in TKS WRKY promoter domains, suggesting being involved in drought-inducibility and flavonoid biosynthetic gene regulation (Figure 6).

WRKY genes play a critical role in plant development. As reported, AtR8 lncRNA defective mutants and *ATWRKY53* and *ATWRKY70* respond to the induction of exogenous SA. The experimental results revealed that the *wrky53* and *wrky70* mutant lines had longer roots than the wild type and mutants of SA [58]. In the current study, the orthologous genes of the two WRKY proteins were TkWRKY70 and TkWRKY27, respectively, which were highly expressed in young and mature roots of TKS (Figure 3 and Figure 7a). In particular, TkWRKY27 was expressed in all tissues, suggesting that the function of this gene was activated at any stage of TKS growth. *ATWRKY13* gene have made outstanding contributions in *Arabidopsis* stem development. The mutants of this gene exhibited a weaker stem phenotype, with the number of thick-walled tissue cells, stem diameters, and vascular bundles being reduced [59]. Similarly, the orthologous protein of ATWRKY13, named TkWRKY57, was specifically expressed in young and mature stems (Figure 7a). This research indicated TkWRKY57 may be involved in regulating the growth and development of TKS stems. Moreover, *ATWRKY75* TFs regulate the acquisition of phosphate and root development in *Arabidopsis*. RNAi silencing inhibited the expression of *ATWRKY75*, which made plants susceptible to phosphorus stress. *ATWRKY75* also cause early accumulation of anthocyanins and increase the length and number of lateral roots and root hairs [60]. TkWRKY28 and TkWRKY30, which have high expression levels in roots, were orthologous genes of *ATWRKY75* (Figure 3). These results revealed the significant contribution of TkWRKY28 and TkWRKY30 in the development of roots. Interestingly, TkWRKY28 and TkWRKY30 may be closely related to the amount of latex produced by TKS because of their higher expression in latex (Figure 7a).

In addition to playing a critical role in plant development, WRKY TFs also play an important role in giving plants tolerance to abiotic stresses, including salinity, osmotic, heat, cold, and injury [54,61]. As everyone knows, temperature, osmotic, and saline-alkali are important factors affecting crop growth, so it is necessary to study the tolerance of TKS to abiotic stresses. We planted some TKS1151 in a greenhouse and treated them with various abiotic stresses and hormones (Figure 1). When salinity and osmotic stress act on plants, it will cause the accumulation of ABA content, consequently activating downstream *WRKY* genes. According to reports by Han et al., *ATWRKY18* and *ATWRKY60* in *Arabidopsis* have a positive effect on the ABA sensitivity of plants, inhibiting seed germination and root growth and enhancing the sensitivity of plants to salinity stress; in turn, *ATWRKY40* is antagonistic to *ATWRKY18* and *ATWRKY60*. This effect suggests that *ATWRKY18* is a weak transcriptional activator that can be induced to function under salt stress [62]. In the current study, the homologous protein TkWRKY71 of *ATWRKY18* was induced under salinity stress and its expression level increased (Figure 3 and Figure 8c). Therefore, it can be inferred that TkWRKY71 may also participate in the salt stress response mechanism of plants by regulating the activity of some hormone-like proteins involved in the ABA signal transduction pathway, thus enhancing the salt tolerance of TKS. The *Arabidopsis* TFs *WRKY46*, *WRKY54*, and *WRKY70* are very important in the regulation pathway of brassinosteroids (BRs). The triple mutant has obvious growth defects and is drought tolerant. More importantly, *ATWRKY54* can interact with the BR master regulator BES1 to co-regulate downstream genes. Moreover, *ATWRKY54* is phosphorylated by GSK3-like kinase, which destroys the protein structure and leads to its instability [63]. Similarly, after PEG treatment, the homologous protein TkWKRY27 in TKS can also be induced into expression; it likely can also interact with some unknown binding protein or regulatory factor to regulate the expression of target genes (Figure 3 and Figure 8d). Interestingly, TkWKRY27 was also activated under salinity stress, so there may be a synergistic effect between these two different stresses. As previously reported, *Arabidopsis wrky25wrky26wrky33* triple mutant is extremely sensitive to heat stress. The germination rate of mutant seeds becomes very low under high-temperature treatment. It was demonstrated that these three WRKY TFs are involved in the cooperation between ethylene activating proteins and HSP-related signaling pathways and mediate the heat-stress response [58]. TkWRKY38 was close to *ATWRKY26* in phylogeny, and the expression level of TkWRKY38 was increased after heat-induced and cold stress in TKS (Figure 3 and Figure 8a,b). TkWRKY38 may be involved in the regulation of similar reactions and play a certain role in improving crop heat tolerance. Notably, under the treatment of ABA, MeJA, and SA, the expression of some genes were significantly downregulated, such as *TkWRKY4*, *TkWRKY23*, and *TkWRKY39* (Figure 9a–c). A similar situation was reported in pineapples, where the transcript levels of many AcWRKY genes were downregulated by heat-stress treatment [64]. These results indicate that some WRKY genes were negatively regulated.

Due to the systematicity of genomics technology, the signaling pathway of TKS in response to stress and the glue-producing pathway of glue-producing plants can be studied in a more comprehensive and in-depth way, providing a more reliable theoretical basis and data support for the biosynthesis mechanism of natural rubber. As Lin et al. assembled high-quality genome sketches of TKS using SMRT, the difficulty of cloning and obtaining key genes for natural rubber biosynthesis was greatly reduced, which was very important for further exploring the gene functions of TKS [29]. Several genes of TKS showed many positive characteristics in the process of coping with adversity. For example, *TkUBC2*, an E2 ubiquitin-binding enzyme gene cloned from TKS, was upregulated under salt stress and UV radiation, but downregulated under drought stress and osmotic stress. This suggested that the gene responded to salt, drought, and osmotic stress in TKS. Another study found that the expression of the *TkJAZ9* gene was upregulated by 2.16 times after 6 h of salt stress in 3-month-old TKS, indicating that this gene responded to salt stress in TKS. In addition to the above cases, *TkDREB2* was selected after a differential expression profile analysis was performed on the transcriptomes treated with MeJA. It was proven by an experiment that *TkDREB2* can increase the drought tolerance of transgenic tobacco and actively respond to MeJA signal transduction regulation of tobacco drought resistance. However, there are very few reports on the TKS gene family, and only functional reports of a single TKS gene can be found. Therefore, this study provides a reference for further study of transcriptomics of TKS under stress.

In addition to studying the signaling pathway of TKS in response to stress, there has been a clear pathway for the rubber biosynthesis of TKS. Genes with high expression in roots and rubber are of particular concern, such as *TkHMGR1*, *TkHMGR2*, and 9 *TkSRPP* genes [29]. In this study, the *TkWRKY38*, *TkWRKY63*, *TkWRKY64*, and *TkWRKY71* proteins showed high expression patterns in roots and latex of TKS, which is beneficial to further study the potential relationship between these proteins and root development, especially rubber production (Figure 7a). In conclusion, a comprehensive analysis revealed the potential functional role of TKS WRKY genes to a certain extent, and contributed to the further characterization of the functions of TKS candidate WRKY gene families, and provided new ideas for agronomic genetic improvement and cultivation of high-quality, stress-tolerant, and high-yielding TKS varieties.

## 4. Materials and Methods

### 4.1. Gene Identification

The genome of TKS was downloaded from NGDC Genome Warehouse (https://ngdc.cncb.ac.cn/gwh/ (accessed on 20 September 2021)). A Hidden Markov Model (HMM) profile of the WRKY DNA-binding domain (PF03106) was used to BLASTP. The Pfam databases (http://pfam.sanger.ac.uk/ (accessed on 27 October 2021)) were used to validate putative WRKY proteins. To verify the conservative domain, those potential sequences were further queried in the InterProScan database (http://www.ebi.ac.uk/interpro/search/sequence-search/ (accessed on 3 November 2021)) and SMART databases (http://smart.embl-heidelberg.de/ (accessed on 4 November 2021)). The physicochemical properties of the putative WRKYs, including the number of amino acids (No. of aa), molecular weight (MW), isoelectric point (pI), aliphatic index, instability index, and GRAVY, were obtained by using the ExPasy tools (https://web.expasy.org/protparam/ (accessed on 9 November 2021)). The subcellular location prediction results were obtained by using tools from Cell-PLoc 2.0 (http://www.csbio.sjtu.edu.cn/bioinf/Cell-PLoc-2/ (accessed on 15 November 2021)).

### 4.2. Sequence Analysis and Alignment

ClustalX software (version 1.81, Desmond G. Higgins and Paul Sharp, Dublin, Ireland) was used to perform multi-sequence alignment on the TKS WRKY protein sequence domain with the default settings, and the results were saved in MSF file format. The core sequences were then manually adjusted and colored. The intron-exon structures of the TKS *WRKY* genes were analyzed by the online tool Gene Structure Display Server (GSDS, http://gsds.gao-lab.org/index.php/ (accessed on 1 December 2021)) [65]. The MEME online program for protein sequence analysis (http://meme.nbcr.net/meme/intro.html/ (accessed on 4 December 2021)) was used to identify the conserved motifs in the identified TKS WRKY protein [66].

### 4.3. Phylogenetic Analysis

MAFFT version 7 (https://mafft.cbrc.jp/alignment/server/ (accessed on 8 December 2021)) was used to perform multi-sequence alignment on the TKS WRKY protein sequence domain with G-INS-1 (progressive method with an accurate guide tree) settings. The phylogenetic tree was constructed by MEGA software (version 7.0, Mega Limited, Auckland, New Zealand) with the neighbor-joining (NJ) method based on the comparison between TKS and *Arabidopsis* proteins; then we used Illustrator software (version 2019, Adobe, San Jose, CA, USA) to beautify and add color. The sequences of the WRKY protein from *Arabidopsis*, maize [50], rice [61,67], grape [47], *Brachypodium distachyon* [68], pineapple [64], peach [69], and poplar [36] were obtained according to the corresponding literature and instructions downloaded from the Phytozome databases (https://phytozome.jgi.doe.gov/ (accessed on 27 December 2021)).

### 4.4. Analysis of Cis-Acting Elements in TkWRKY Promoter Regions

The upstream sequences (2000 bp) of the TKS were obtained from NGDC Genome Warehouse. PlantCARE (http://bioinformatics.psb.ugent.be/webtools/plantcare/html/ (accessed on 5 January 2022)) was utilized to identify the twelve *cis*-elements of the TkWRKY promoter regions [70].

### 4.5. Plant Materials and Treatments

The dandelion line 1151 was planted in the greenhouse of Anhui Agricultural University, Hefei, China (31°87′ N 117°26′ E). TKS1151 was grown in 16/8 h light/dark at 22 ± 2 °C and 76% humidity. The seeds were germinated and grown in black soil and vermiculite mixture in a ratio of 1:1. TKS at the seven-leaf stage were subjected to 40 °C and 4 °C for heat and cold treatments, respectively, and the leaves were collected at 3, 6, 9, 16, and 24 h for both treatments. For salinity and osmotic treatments, seedlings were sprayed with 150 mM NaCl and 15% PEG6000 solutions for 3, 6, 9, 16, and 24 h. Finally, the leaves were sprayed with 300 μM ABA, 100 μM SA, and 50 μM MeJA for hormone treatment; all hormones were dissolved in sterile water [53], and sterile water was sprayed on leaves from the same batch as the negative control. In contrast, leaves were collected at 3, 9, 16, 24, and 48 h after treatments.

### 4.6. RNA Extraction and Real-Time RT-PCR Analysis

Total RNA was extracted using the Trizol method [71,72]. All samples were tested for the concentration, A260/A280 ratio, and A260/A230 ratio by a Thermo SCIENTIFIC spectrophotometer. HiScript^®^ III RT SuperMix (Vazyme, Nanjing, China) was used to remove residual genomic DNA contamination and reverse transcription to obtain cDNA for qPCR. The Real-Time RT-PCR was carried out with a Roche Lightcyler^®^ 480 instrument using SYBR qPCR Master Mix in 20 μL volumes. The stable and reliable housekeeping TKS *β-actin* gene and *GAPDH* gene were used as internal controls. The Real-Time RT-PCR primers were designed by Primer Premier software (version 5, Premier, Vancouver, BC, Canada) (Appendix A). The standard amplification procedure of the Real-Time RT-PCR was as follows: first, the holding stage is the reaction at 95 °C for 30 s, followed by the cycle stage, which runs 40 cycles at 95 °C for 5 s and 60 °C for 34 s, and finally the melting curve stage. The expression level was expressed as a change in relative multiples of 0 h (control) set to 1. Each reaction was performed in three biological replicates and the Real-Time RT-PCR data were analyzed by 2^−ΔΔCT^ method. The bar graph was drawn by GraphPad Prism (version 8.0.2, GraphPad Software, San Diego, CA, USA).

### 4.7. Expression Profiling of TkWRKYs in Different Tissues Based on the Public RNA-seq Data Sets

RNA-seq data sets of 12 different TKS tissues, including flower (FL), latex (LA), peduncle (PE), seed (SE), mature leaf (ML), mature lateral root (MLR), mature main root (MMR), mature stem (MS), young leaf (YL), young lateral root (YLR), young main root (YMR), and young stem (YS), were obtained from NGDC Genome Warehouse. The gene expression level was obtained by FPKM value [73]. A heat map was obtained from HemI1.0 and generated based on a log_2_ (FPKM + 1) value conversion [74]. The boxplot was drawn using Origin software (version 2018, OriginLab, Northampton, MA, USA).

## 5. Conclusions

In this study, 72 genes encoding WRKY TFs in *Taraxacum kok-saghyz* Rodin (TKS) were identified. TkWRKYs were divided into three main populations. Many identified WRKY proteins have close evolutionary relationships with *Arabidopsis* proteins, suggesting possible similarity of WRKY evolutionary patterns in dicotyledons. The WRKY domain is conserved in TKS, and the gene structure and motif analyses showed that the same group contained a highly similar exon-intron structure and motif. The analysis of *cis*-elements of the TKS promoter gene provides valuable clues for understanding the evolutionary characteristics of the TKS WRKY gene. According to the tissue-expression pattern and abiotic-stress quantitative expression results, WRKY genes play important roles in TKS development and stress resistance, such as cold-stress response (*TkWRKY18*, *TkWRKY23*, and *TkWRKY38*), heat-stress response (*TkWRKY21* and *TkWRKY38*), salt-stress response (*TkWRKY71*), and osmotic stress (*TkWRKY27*). However, whether these WRKY genes can participate in the plant hormone signal transduction pathway or interact with other TFs in TKS needs further study.

## Figures and Tables

**Figure 1 ijms-23-10270-f001:**
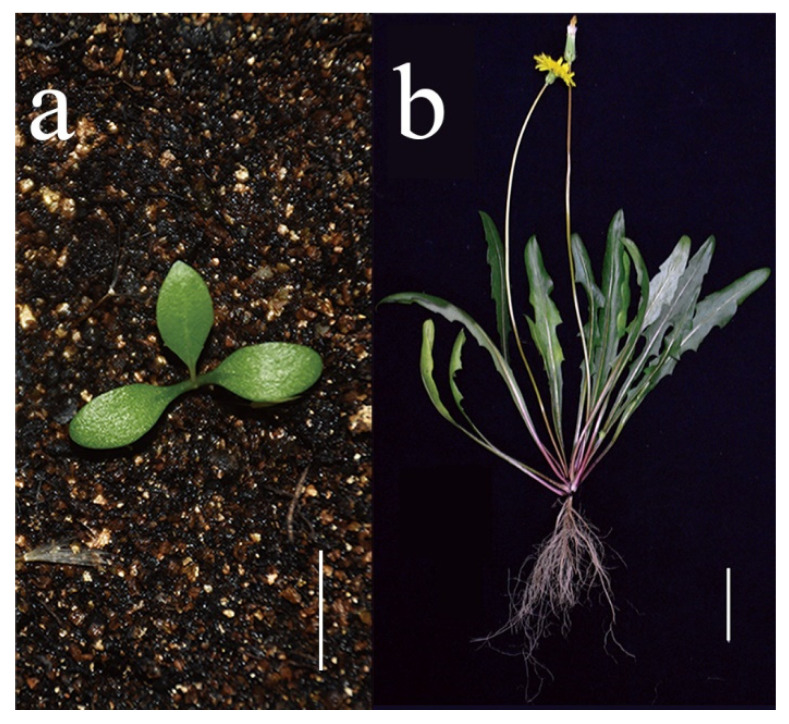
Morphologies of *Taraxacum kok-saghyz* at different growth stages. (**a**) Morphologies of *Taraxacum kok-saghyz* at young leaves stage. Bar = 1 cm. (**b**) Morphologies of *Taraxacum kok-saghyz* at mature leaves stage. Bar = 5 cm.

**Figure 2 ijms-23-10270-f002:**
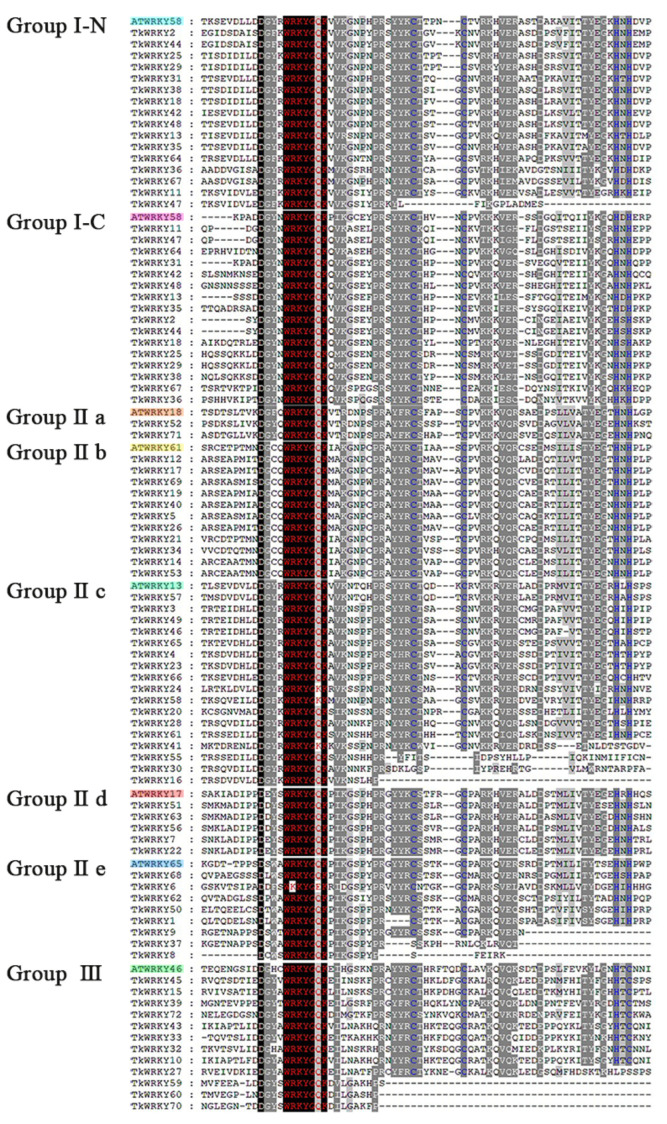
Domain recognition in the TKS WRKY protein sequences. The ATWRKY domain is color-coded.

**Figure 3 ijms-23-10270-f003:**
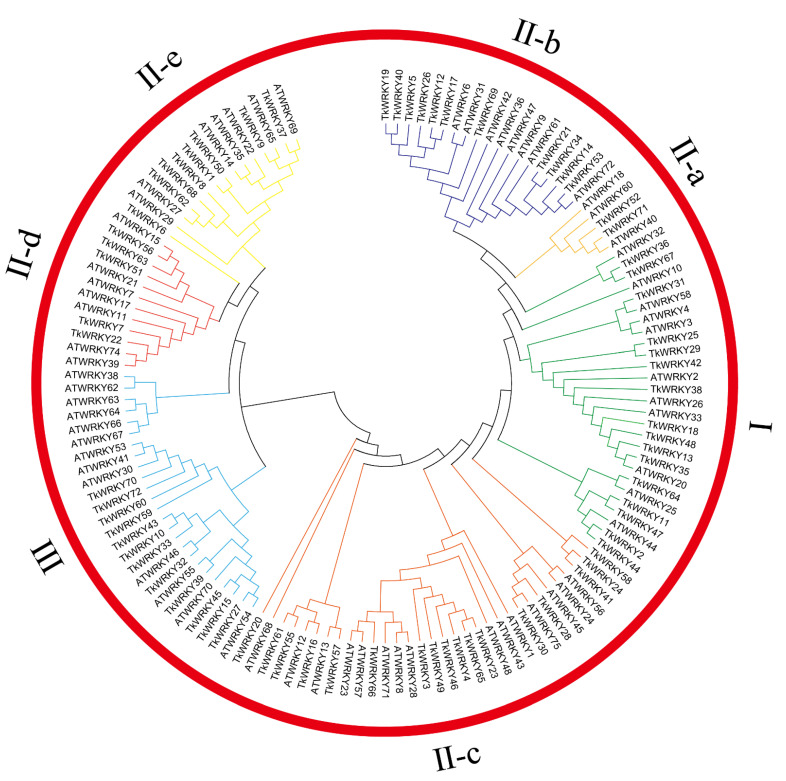
The phylogeny of *Arabidopsis* and *Taraxacum kok-saghyz* Rodin TFs.

**Figure 4 ijms-23-10270-f004:**
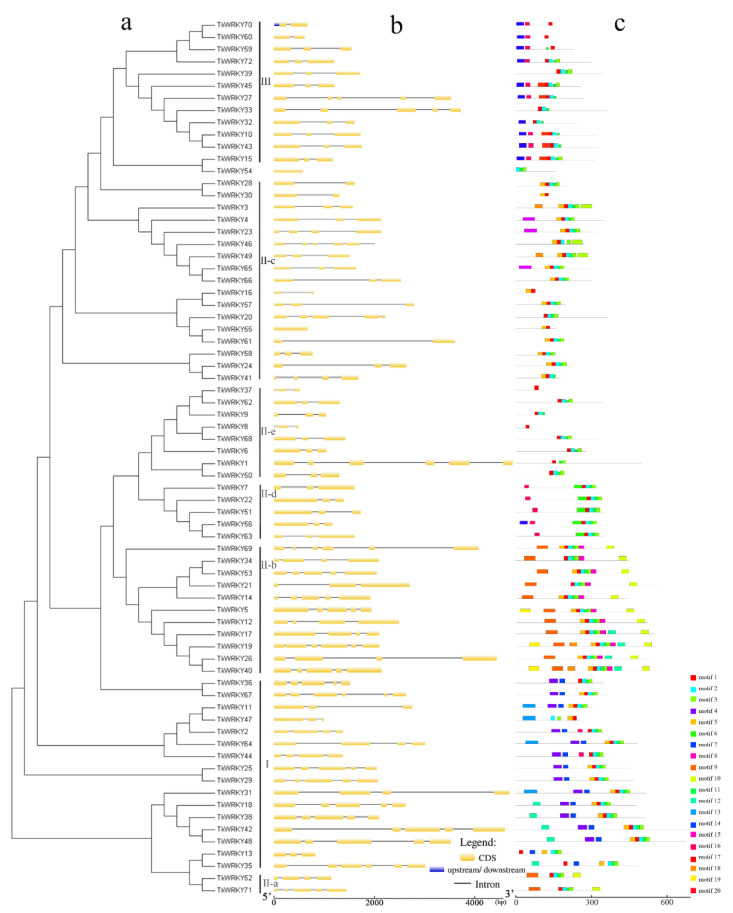
Phylogeny (**a**), gene structure (**b**), and putative motif analysis (**c**) of each TkWRKY gene and WRKY protein. Exons, introns, and upstream/downstream are represented by yellow, black, and blue lines, respectively. The length and proportion are measured with a ruler below. The length and position of the 20 motifs are represented by differently colored boxes.

**Figure 5 ijms-23-10270-f005:**
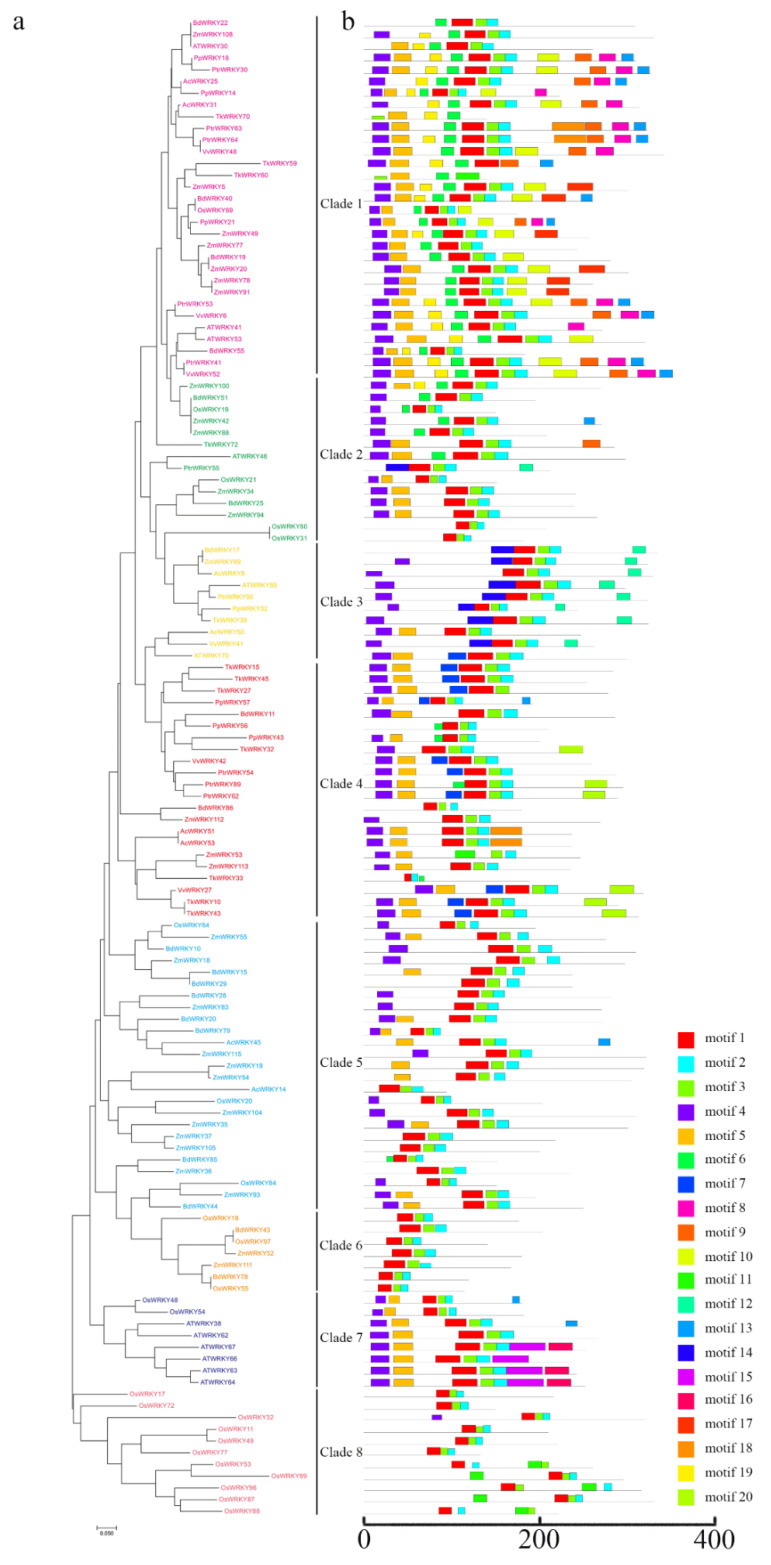
Phylogenetic relationships (**a**) and putative motifs (**b**) of Class III WRKY proteins in nine plants. The proteins are grouped into eight branches. The length and position of the 20 motifs are represented by differently colored boxes.

**Figure 6 ijms-23-10270-f006:**
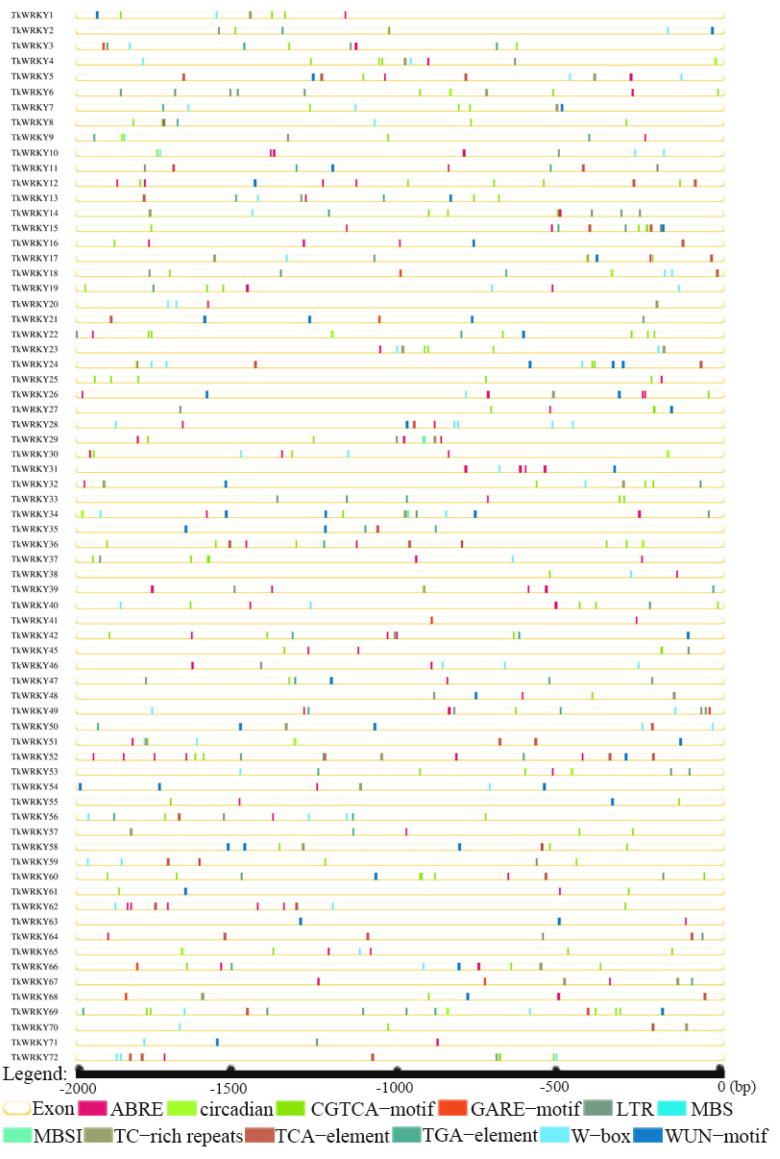
Location distribution of different types of *cis*-elements in the TkWRKY promoter region; the 12 differently colored lines correspond to 12 different *cis*-elements.

**Figure 7 ijms-23-10270-f007:**
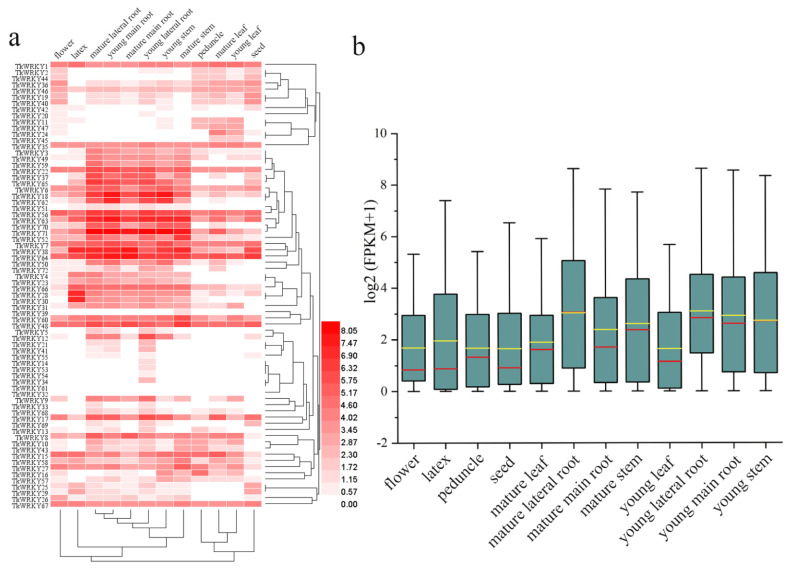
Expression profiling of TKS WRKY genes in 12 tissues of flower (FL), latex (LA), peduncle (PE), seed (SE), mature leaf (ML), mature lateral root (MLR), mature main root (MMR), mature stem (MS), young leaf (YL), young lateral root (YLR), young main root (YMR), and young stem (YS): (**a**) phylogenetic relationship clustering of the expression profiles of the WRKY genes from 12 tissues; (**b**) boxplots showing the dispersion of the WRKY genes expression profile in 12 tissues. Red and yellow represent the median line and average line, respectively.

**Figure 8 ijms-23-10270-f008:**
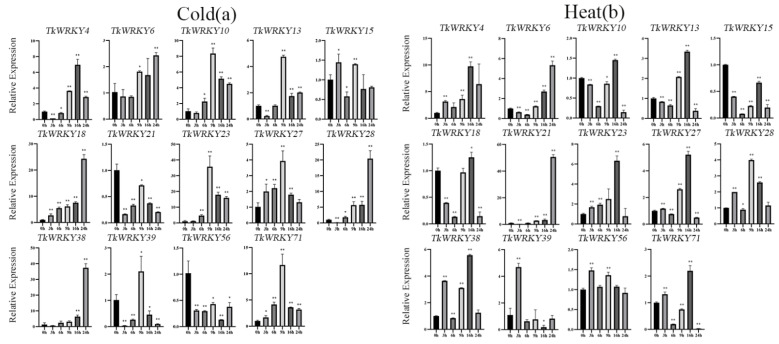
Modulation of gene expression of 14 TkWRKY genes at 0 (control), 3, 6, 9, 16, and 24 h after (**a**) cold, (**b**) heat, (**c**) salinity, and (**d**) osmotic stress. Data were normalized to *β-actin* and glyceraldehyde-3-phosphate dehydrogenase (*GAPDH*). Vertical bars indicate the standard deviation whereas an asterisk indicates the summary of the *p* value of the independent sample T-test of the corresponding gene compared with the control (* *p* < 0.05, ** *p* < 0.01).

**Figure 9 ijms-23-10270-f009:**
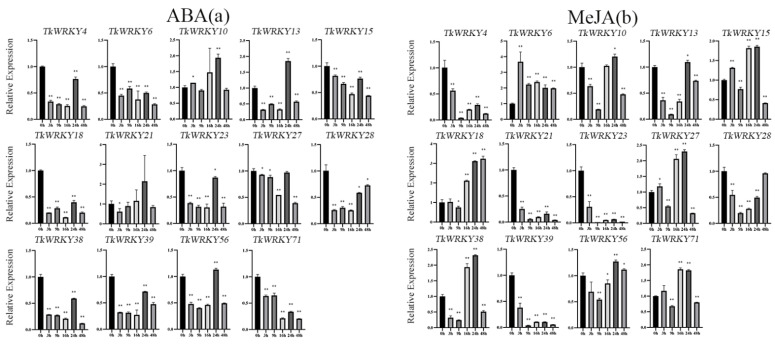
Modulation of gene expression of 14 TkWRKY genes at 0 (control), 3, 9, 16, 24, and 48 h after (**a**) ABA, (**b**) MeJA, and (**c**) SA treatments. Data were normalized to *β-actin* and *GAPDH*. Vertical bars indicate the standard deviation whereas an asterisk indicates the summary of the *p* value of the independent sample T-test of the corresponding gene compared with the control (* *p* <0.05, ** *p* < 0.01).

## Data Availability

All data are displayed in the manuscript.

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
