# Peer review of "Identification of the WRKY Gene Family and Characterization of Stress-Responsive Genes in Taraxacum kok-saghyz Rodin"

_ijms, 2022, doi:10.3390/ijms231810270_

Round 1

Reviewer 1 Report

Article written in clear English and with an introduction full of bibliographical references. Numerous and valid materials and methods to demonstrate the experimental approach. The figures and tables are excellent and descriptive. Clear results, with many references to what is reported in literature. Conclusions in accordance with what they declare in the experimental design but basic research to be expanded above all to better specify the declared practical advantage of using Taraxacum kok-saghyz Rodin as an alternative source of natural rubber production.

You could add some images of the TKS in the introduction

Line 29 insert this is, because it is hard to understand

TkWRKY33 and TkWRKY43 are missing in the phylogenetic tree of figure 2

Figures 6a and b not shown

Line 438 unclear, “the induction of exogenous SA. The experimental results revealed that the wrky53 and ”, put, instead of.

Author Response

Dear reviewer #1:

Many thanks for your great efforts on processing our manuscript (Manuscript ID ijms-1862628, Identification of WRKY gene family and characterization of stress-responsive genes in Taraxacum kok-saghyz Rodin). We are sincerely grateful to your comments and suggestions for improving the manuscript, which really help us greatly to improve the quality of our manuscript. Based on your suggestions, we have made careful revisions in this submitted manuscript and added some information to different parts of the manuscript according to your suggestions. We wish you are satisfied with the revised manuscript.

Reviewer 2 Report

The paper “Identification of WRKY gene family and characterization of stress-responsive genes in Taraxacum kok-saghyz Rodin” by Yifeng Cheng, Jinxue Luo, Hao Li, Feng Wei, Yuqi Zhang, Haiyang Jiang, and Xiaojian Peng is devoted to investigation of WRKY gene family in the genome of the Taraxacum kok-saghyz Rodin and the possible role of this gene family as transcription factors in the transmission of stress signals in higher plants.

The authors have carried out a thoughtful study, the results of the study are convincing and complete. The paper deserves the publication and represents a new knowledge about WRKY gene family in T. kok-saghyz.

However, I have a number of comments on the presented paper.

My main concern is the way of presenting the Discussion section. This section looks like a story about WRKY gene family instead of being a discussion of the results obtained in the present study with references to published studies. The Discussion section, although it contains the words "in this study", I find it difficult even to guess how to insert references to the Figures of the present study. Please modify this section so that it actually discusses all the most important results of your research with references to all the Figures presented.

 Other comments:

1.     There is no logical connection between the first sentence of the Abstract and the rest text in this section. Please change this.

2.     L.100. The abbreviation GRAVY in the text should be introduced at the first mention, whereas it is introduced in L.533.

3.     The quality of some Figures (1, 3, 4 and 5) should be improved. Some symbols in these figures are illegible.

4.     L.252. The term “cis-elements” occurs for the first time. Please add explanatory information about these elements in the introduction or in the results just before reporting the results concerning the quantitative and functional analysis of cis-elements in TkWRKY promoter regions. Not every reader may know that these are the regions of DNA or RNA that regulate gene expression, which are the binding sites for one or more transcription factors.

5.     L.266-279. This part of the Results looks like a part of Introduction or Discussion. In fact, it's great if authors include some short conclusions of the results after the Results sections, but in this case, data from other authors is discussed here. Please, move this part to another section of the manuscript.

6.     Please, check the order of Supplementary Tables. They are mentioned for the first time in surprising way in the text: Table S1 is in the L.104, Table S2 is in the L.577, Table S3 is in the L.257, Table S4 is in the L.218, Table S5 is in the L.249.

7.     In the page 359 there is a Figure without a number, a legend and without a mention in the manuscript.

8. L. 371. The abbreviation GAPDH in the text should be introduced at the first mention.

Author Response

Dear reviewer #2:

Many thanks for your great efforts on processing our manuscript (Manuscript ID ijms-1862628, Identification of WRKY gene family and characterization of stress-responsive genes in Taraxacum kok-saghyz Rodin). We are sincerely grateful to your comments and suggestions for improving the manuscript, which really help us greatly to improve the quality of our manuscript. Based on your suggestions, we have made careful revisions in this submitted manuscript and added some information to different parts of the manuscript according to your suggestions. We wish you are satisfied with the revised manuscript.

Round 2

Reviewer 2 Report

Dear authors, despite the assurances in the response to the Reviewer that the shortcomings of the text according to the comments were corrected, the most significant problems were not eliminated. In addition, there are new mistakes in the new version of Manuscript. I am attaching a file with my responses to your responses to the points where problems still exist.

Author Response

Dear reviewer #2:

Thank you very much for your critical review of our manuscript. We are so sorry to make these mistakes, a carefully revised manuscript was submitted, and a lot of revised information was not listed here. Thank you again for your careful correction and guidance, especially you took the trouble to illustrate the problem for us.

Round 3

Reviewer 2 Report

The authors have corrected the MS according to all of my suggestions.